# A Classification Approach for Cancer Survivors from Those Cancer-Free, Based on Health Behaviors: Analysis of the Lifelines Cohort

**DOI:** 10.3390/cancers13102335

**Published:** 2021-05-12

**Authors:** Francisco O. Cortés-Ibañez, Sunil Belur Nagaraj, Ludo Cornelissen, Grigory Sidorenkov, Geertruida H. de Bock

**Affiliations:** 1Department of Epidemiology, University Medical Center Groningen, University of Groningen, 9713 GZ Groningen, The Netherlands; g.sidorenkov@umcg.nl (G.S.); g.h.de.bock@umcg.nl (G.H.d.B.); 2Department of Clinical Pharmacy & Pharmacology, University Medical Center Groningen, University of Groningen, 9713 GZ Groningen, The Netherlands; sbn1984@gmail.com; 3Department of Radiation Oncology, University Medical Center Groningen, University of Groningen, 9713 GZ Groningen, The Netherlands; l.j.cornelissen@rug.nl

**Keywords:** cancer survivors, health behaviors, lifestyle, classification, machine learning, medical informatics

## Abstract

**Simple Summary:**

Health behaviors affect health status in cancer survivors. We aimed to identify such key health behaviors using nonlinear algorithms and compare their classification performance with logistic regression, for distinguishing cancer survivors from those cancer-free in a population-based cohort. We used health behaviors and socioeconomic factors for analysis. Participants from the Lifelines population-based cohort were binary classified as cancer survivors or cancer-free using nonlinear algorithms or logistic regression. Data were collected for 107,624 cancer-free participants and 2760 cancer survivors. Using all variables, algorithms obtained an area under the receiver operator curve (AUC) of 0.75 ± 0.01. Using only health behaviors, the algorithms differentiated cancer survivors from cancer-free participants with AUCs of 0.62 ± 0.01 and 0.60 ± 0.01, respectively. In the case–control analyses, both algorithms produced AUCs of 0.52 ± 0.01. The main distinctive classifier was age. No key health behaviors were identified by linear and nonlinear algorithms to differentiate cancer survivors from cancer-free participants.

**Abstract:**

Health behaviors affect health status in cancer survivors. We hypothesized that nonlinear algorithms would identify distinct key health behaviors compared to a linear algorithm and better classify cancer survivors. We aimed to use three nonlinear algorithms to identify such key health behaviors and compare their performances with that of a logistic regression for distinguishing cancer survivors from those without cancer in a population-based cohort study. We used six health behaviors and three socioeconomic factors for analysis. Participants from the Lifelines population-based cohort were binary classified into a cancer-survivors group and a cancer-free group using either nonlinear algorithms or logistic regression, and their performances were compared by the area under the curve (AUC). In addition, we performed case–control analyses (matched by age, sex, and education level) to evaluate classification performance only by health behaviors. Data were collected for 107,624 cancer free participants and 2760 cancer survivors. Using all variables resulted an AUC of 0.75 ± 0.01, using only six health behaviors, the logistic regression and nonlinear algorithms differentiated cancer survivors from cancer-free participants with AUCs of 0.62 ± 0.01 and 0.60 ± 0.01, respectively. The main distinctive classifier was age. Though not relevant to classification, the main distinctive health behaviors were body mass index and alcohol consumption. In the case–control analyses, algorithms produced AUCs of 0.52 ± 0.01. No key health behaviors were identified by linear and nonlinear algorithms to differentiate cancer survivors from cancer-free participants in this population-based cohort.

## 1. Introduction

In Europe, the number of new cancer diagnoses increased from 3.2 million people in 2008 to 3.9 million people in 2018 [1]. Around half of these are now expected to survive longer than 10 years [2], with estimates for Europe indicating that the 8.7 million cancer survivors in 2007 had exceeded 10 million by 2012 [3]. The worldwide number of cancer survivors was estimated to be close to 44 million in 2018 [4]. This increased survival can be explained by the greater number of patients diagnosed with cancer. This increase is mainly due to population aging, as age is considered the main risk for cancer [5]; however, enhancements in early-detection programs, staging methods, and the effectiveness of treatments have also contributed [6]. In light of the increasing importance of cancer survivor prognosis, the relatively sparse research done in the field of cancer survivors leads to the pursuit of investigations to evaluate, understand, and improve their health outcomes (i.e., lifestyle and psychosocial well-being) [7]. Special attention should be focused on health behaviors that are prone to affect prognosis after cancer has been diagnosed, to reduce the risk of recurrence or a second malignancy [7]. Conversely, recent literature suggests that cancer survivors only improve their health behaviors in the short term postdiagnosis, but in the long term return to what they were used to, either risky or unhealthy behaviors [8].

The main health behaviors that are widely associated with an increase in the risk of cancer or cancer recurrence include smoking [9], alcohol consumption [10], unhealthy diet [11,12], low physical activity levels [13], high body mass index (BMI) [14], and a more sedentary behavior [15]. Socioeconomic factors like age [16], sex [17], and education level [18] may also have a central role in the risk of cancer. Importantly, the assessment and identification of health behaviors in cancer survivors compared to cancer-free individuals could help them to be aware of their health behaviors and strengthen their prognosis [7]. Studies have evaluated/identified the differences in health behaviors of cancer survivors compared to cancer-free people in population-based studies, most commonly using techniques such as logistic regression or prevalence ratios [19,20,21,22,23,24,25,26,27]. The most frequently evaluated health behaviors in cancer survivors are alcohol consumption, smoking, and physical activity [19,20,21,22,23,24,25,26,27]. However, not all the studies had the possibility to evaluate additional variables such as diet [19,20,22,27], BMI [21,22], or sedentary behavior [15]. In addition, these studies were performed by independently evaluating the associations between health behaviors and cancer recurrence, and typically adjusted for age and sex [28]. In this way, such research lacked the ability to assess complex or nonlinear relations among the included characteristics, or had a relatively small sample size [29,30]. Additionally, there is evidence suggesting that nonlinear methods should be used to identify relevant associations between variables and the outcome in aging-related research; therefore, those methods should be implemented and compared with traditional approaches [31]. In particular, as lifestyle in cancer survivors is getting more attention due to the increased rates mentioned before, the nonlinear methods might show better performance in modeling the relationship between health behavior and cancer recurrence [28]. Some recent studies have evaluated the performance of supervised machine-learning methods, for the classification and/or prediction of cancer prognosis and identification of key variables [32,33,34,35]. However, these methods have not been applied to identify possible differences in health behaviors between cancer survivors and those without cancer in a representative population-based cohort, nor compared the performance to those commonly used linear approaches. Those might evaluate if there are linear, nonlinear, or complex associations between the variables and the outcome. In addition, the comparison of the performance of several algorithms would provide a better scenario for such evaluation [31]. We hypothesized that a nonlinear algorithm could better classify participants as cancer survivors or cancer-free by identifying distinct key health behaviors and socioeconomic factors compared with a traditional linear algorithm. Thus, our aims were to: (i) use the nonlinear algorithms to identify the key health behaviors in cancer survivors in a population-based cohort, and (ii) compare the performances of linear and nonlinear algorithms in classifying participants as cancer survivors or cancer-free based on their health behaviors and socioeconomic factors.

## 2. Materials and Methods

### 2.1. Study Design

This cross-sectional study used baseline data from the prospective, population-based Lifelines cohort [36]. Lifelines used a three-generation design to include a representative sample of 10% of people living in the north of the Netherlands, with 167,729 participants aged 6 months to 93 years recruited from 2006 to 2013. Participants provided information about their lifestyles, socioeconomic statuses, and cancer histories (if present), among other details, via a self-administered questionnaire. The original study protocol was approved by the Medical Ethics Review Committee of the University Medical Center Groningen [36].

### 2.2. Participants

Our analysis included adult participants (age ≥ 18 years) with complete data for three socioeconomic factors (i.e., age, sex, and educational level) and six health behaviors (i.e., BMI, smoking, alcohol intake, physical activity, diet quality, and sedentary behavior). Those cases with self-reported skin cancer were excluded because it can be over-reported [37]. A total of 110,384 participants were included, as detailed in Figure 1.

### 2.3. Measurements of Health Behaviors and Socioeconomic Factors

Cancer was considered present when participants answered affirmatively to the question ‘Have you ever been diagnosed with cancer?’ (i.e., self-reported) and provided both their age at diagnosis and the type of cancer. The time since diagnosis was then estimated based on the difference between the age at diagnosis and the age at data collection and dichotomized as ≤5 years and >5 years. Socioeconomic status was indicated by the level of education, which is commonly used for this purpose because it is easy to measure, can be self-reported, and correlates well with other indicators of social stratification, such as income. By contrast, income is often considered to be private by study participants and is therefore more likely not to be reported [38]. Education was classified as low, medium, or high.

Since height and body weight were assessed at several Lifelines research sites, BMI was included as provided by their own calculations (weight divided by the square height (kg/m^2^) of the participants).

Smoking was considered in two ways. First, we calculated the total grams of tobacco currently smoked, using the following equivalence: 1 cigarette = 1 g, one cigarillo = 3 g, and one cigar = 5 g [39]. Second, smoking was expressed as never (those who smoked for less than 1 year), former (smoking more than 1 year but already stopped for 1 month previous to data collection), and current smoking (participants who reported smoking during the last month).

Alcohol intake was assessed by a 110-item food-frequency questionnaire. We considered the total amount of alcohol in grams/day that a participant reported in the previous month. Every standard drink reported contained around 10 g of alcohol, according to Dutch dietary guidelines [40].

Physical activity was measured by a Dutch validated questionnaire (Short Questionnaire to Assess Health-Enhancing Physical Activity-SQUASH) [41]. From this evaluation, we only considered moderate to vigorous activities. The threshold to be relevant in health was a minimum of 150 min of moderate to vigorous physical activity per week [42].

Diet quality was assessed with the 110-item food-frequency questionnaire that measured food intake over the previous month. A food-based Lifelines Diet Score (LLDS) was constructed based on the food-frequency questionnaire scores [43]. The LLDS incorporates the latest guidelines for healthy food and indicates the relative intake of food by known positive or negative health effect. The LLDS ranges from 0 (lowest diet quality) to 48 (highest diet quality) [43]. Each point increase in the score represents a shift of one quintile in one of the food groups in a healthier way.

Sedentary behavior was evaluated by the total number of hours the participant spent watching TV per day. The participants answered the following questions: ‘On average how many hours per day do you spend watching TV?’ and ‘On average how many minutes per day do you spend watching TV?’. The total amount of hours/day per participant was calculated by adding the two values.

### 2.4. Statistical Analysis

The data set was highly imbalanced (2760 cancer survivors and 107,624 cancer-free participants), which can severely bias the performance of machine-learning algorithms [44]. To account for this, we used a sample-size equalization method that involved randomly grouping participants with no history of cancer into 39 equal subsets based on the number of cancer survivors (107,624/2760 = 39 subsets). This balanced the distribution of participants in the two groups, as shown in Figure 2.

For each subset, we performed supervised binary classification between the cancer survivors and participants with no history of cancer using fivefold cross-validation. During classification, we randomly used 80% of the data for training and the remaining 20% for testing. Continuous variables were then normalized in the training set by using uniform means and standard deviations (subtracted from the mean and divided by the standard deviation); in the testing set, this was by the mean and standard deviation of the training set. This procedure resulted in a total of 39 binary classification models corresponding to each subset.

Three different nonlinear algorithms were used for binary classification. The first one was a random forest, in which we tuned three hyperparameters: (i) number of trees starting from 100 and increasing by 100 until 500 trees, and since area under the receiver operator curve (AUC) did not increase from 300 to 500 trees, the later one was set for all the models; (ii) the number of variables selected for each split (mtry) was set to 1–4 in each split; and (iii) the node size. The mean decrease in the Gini index (MDG) was used to evaluate the importance of health behaviors and socioeconomic factors. The second algorithm was a support vector machine, in which the values of the hyperparameter “C” (cost of constraint) were set for a search (0.25, 0.5, 1). The third one was a gradient boosting machine, and the following hyperparameters were tuned: (i) eta 0.3, 0.5; (ii) gamma (0, 0.01); and (iii) max depth (1, 4, 6). The AUC was used as a performance metric (see Figure 2).

Finally, a case–control analysis was performed using only the health behaviors, in which we matched cancer survivors to cancer-free participants (1:1) by socioeconomic factors (i.e., age, sex, and education level) before classification.

We compared the performances of the nonlinear algorithms (i.e., random forest, support vector machine and gradient boosting machine) with that of a traditional logistic regression model. All analyses were performed in R Statistics (Version 3.5.2) with the ‘Caret’ package. All results are reported as mean AUC ± standard deviation unless stated otherwise.

## 3. Results

### 3.1. Descriptive Analysis of the Cohort

In the final data set, females accounted for 58.7% (*n* = 64,793), the median age was 44 years (interquartile range (IQR) = 16), and 68.7% (*n* = 75,854) reported a low or medium educational level (Table 1). Only 2.5% (*n* = 2760) of the cohort reported a history of cancer, and about half of these (58.3%; *n* = 1607) had been diagnosed for >5 years by the time they completed the questionnaires. The most common self-reported cancers were breast (36%; *n* = 986), endometrium (16%; *n* = 441), bowel (8.5%; *n* = 233), and prostate (7.61%; *n* = 210). Most cancer survivors were female (68.2%), had a median age of 57 years (IQR = 18), had a low education level (43.8%), and were former smokers (46.7%).

### 3.2. Performance of Individual Variables in Differentiating Cancer Survivors from Cancer-Free Participants

Prior to any other analysis, we evaluated the individual performances of socioeconomic factors and health behaviors at differentiating cancer survivors from cancer-free participants. The highest AUC was obtained for age (0.74 ± 0.01), with all other factors having AUCs below 0.60: sex, 0.55 ± 0.01; education level, 0.57 ± 0.01; BMI, 0.56 ± 0.02; alcohol intake, 0.52 ± 0.02; smoking, 0.51 ± 0.01; physical activity, 0.51 ± 0.01; sedentary behavior, 0.58 ± 0.01; and diet, 0.58 ± 0.01.

### 3.3. Performance of Nonlinear Algorithms

We then used the nonlinear algorithms, including all health behaviors and socioeconomic factors, to differentiate cancer survivors from people with no history of cancer. For the random forest, this produced an overall mean AUC of 0.75 ± 0.01; the support vector machine had an overall AUC of 0.76 ± 0.02, and the gradient boosting machine an AUC of 0.74 ± 0.01 (see Table 2, Appendix A). Age remained the dominant predictor (see Table 3, Appendix A).

The mean AUC dropped for all algorithms to ≤0.66 after excluding age from the prediction, and it dropped slightly further when excluding age and sex or age and educational level (Table 2). The health behaviors with higher scores in the Gini index were BMI (MDG, 100), alcohol consumption (MDG, 99.42), and physical activity (MDG, 83.23) (Table 3).

### 3.4. The Case–Control Analysis

Since the ratio of participants in this specific part of the analysis was 1:1 (exact match for age, sex, and education level), this section shows the overall results obtained only from the classification performance of health behaviors (see Table 2 and Table 3). Logistic regression resulted in an overall AUC of 0.52 ± 0.01, the support vector machine had an AUC of 0.55 ± 0.01, the gradient boosting machine had an AUC of 0.53 ± 0.01, and the random forest algorithm’s AUC was 0.52 ± 0.01. This last one revealed that BMI (MDG, 100), alcohol consumption (MDG, 99.15), physical activity (MDG, 84.93), and diet (MDG, 76.93) were the health behaviors with higher scores in the Gini index (Table 3). The nonlinear algorithms had a comparable prediction performance to that of a traditional logistic regression algorithm, as summarized in Table 2 (also see Appendix A).

## 4. Discussion

In this study, our aims were to use supervised nonlinear algorithms (i.e., random forest, support vector machine, and gradient boosting machine) to identify key health behaviors in cancer survivors, and to compare the classification performance of linear and nonlinear algorithms when differentiating cancer survivors and cancer-free participants based on health behaviors and socioeconomic factors. Both the linear and nonlinear algorithms provided similar results in the case–control analysis when classifying participants as cancer survivors or cancer-free based only on their health behaviors, giving an overall AUC of <0.55. Our results were in line with a recent systematic review that showed no differences in the performance of machine-learning algorithms over logistic regression in clinical prediction scenarios in which the type of data analyzed is commonly tabular or the amount of variables included is relatively small [45]. Results obtained in this study suggested that differences among health behaviors in the Dutch Lifelines cohort were not relevant enough to make a proper classification of cancer survivors and cancer-free participants, and this may be due to the relatively homogenous distribution of such health behaviors among this population. This is in agreement with a previous approach in this cohort that used only logistic regression and did not identify substantial differences among cancer survivors and the general population [46]. However, also consistent with previous research [16], age was the dominant predictor (0.74 ± 0.01): after excluding age from the prediction, cancer survivors and participants with no history of cancer did not differ considerably by any other health behavior.

The role of BMI in cancer survivors is controversial. There is evidence that increased BMI is associated with an increased risk for some types of cancer [14], so one might also expect that cancer survivors would have an increased BMI, but our results are not sufficient to support those findings. However, our findings are more similar to other studies reporting no relevant differences between cancer survivors and the general population [19,24,25,26], and contrast with other research indicating that cancer survivors have a marginally higher BMI (odds ratio, 1.19; 95% confidence interval, 1.01–1.39) [20]. An explanation for these conflicting results may be that the cohorts included different cancers, but we have no data to support this explanation.

Many studies have indicated that cancer survivors are more likely to be former smokers [19,21,22,23,47], but some have found no substantial differences in current smoking status among cancer survivors and the general population [24,25]. In our study, the percentage of current smokers was 16.6% among the cancer survivors, comparable with data published elsewhere [8]. However, the percentage was higher in the population-based cohort (20.4%) when compared with the cancer survivors (16.6%) and the age-matched cohort (15.4%), suggesting that younger age might be the main determinant of being a current smoker. Regarding alcohol consumption, several previous population-based studies have shown that cancer survivors are less likely to be alcohol users [19,21,23,24]. By contrast, other studies have reported comparable alcohol consumption to the general population [20,22,26], as shown by our results.

Similarly, several population-based studies have reported no substantial differences in diet between cancer survivors and those with no history of cancer [19,20,22,25,26], and other studies have not been able to evaluate diet in their analyses [21,23,24]. Our findings were consistent with those studies concluding that there were no relevant differences in diet for cancer survivors. Some possible explanations are: (i) that participants answered diet questions in a way they considered to be socially desirable; (ii) that the assessments were of poor quality (i.e., only including fruit and vegetable consumption); or (iii) that cancer survivors had readopted a diet similar to their peers.

Whereas some studies reported that cancer survivors were less likely to be physically active compared with the general population [19,22,47], others reported that cancer survivors were more likely to be physically active [23,24]. However, others still have produced results that were similar to ours, indicating that there were no marked differences between groups based on physical activity levels [21,25]. Discrepancies regarding physical activity might be explained by two factors. First, self-reported physical activity is sensitive to over- or underestimation [48]. Second, there may have been differences between the study questionnaires used to collect data.

Because sedentary behavior was not evaluated as a separate factor in previous studies, we are unable to offer a direct comparison. Our motivation to add it as a separate health behavior was that there is evidence that it increases the risk for some types of cancer [15], and that cancer survivors tend to have large periods of sedentary behavior during the day [49], yet it has not previously been considered as an independent factor and compared to a healthy population. This may be due to the limited amount of data collected through questionnaires, or the lack of evidence to motivate the collection of such data. We will continue to consider sedentary behavior separately from physical activity in further analysis, in anticipation that this will either support or refute the existing evidence.

A strength of this study was that we successfully applied a supervised machine-learning algorithm as a methodological approach to identify the most important health behaviors among cancer survivors. It also benefitted from our comparison of the performance of the several models to that of a traditional logistic regression approach. The weighting based on the MDG has added important data about the contribution of each factor for the classification of cancer survivors in a nonlinear model. In addition, all analyses were conducted with complete information for the selected health behaviors obtained from a representative population-based cohort of 110,384 participants (2760 cancer survivors) in the north of the Netherlands, and analyzed not only a subsample, but evaluated the performance of the entire data set while eliminating the class-imbalance issue; in this way, we overcame limitations from previous studies regarding sample size or missing data.

Another key strength is that all health behaviors were comprehensively measured and validated against Dutch guidelines. Diet was not only assessed by the consumption of fruits and vegetables, but for a wide-range of diet scores across nine food groups [43], which is more in line with evidence suggesting that a low-quality diet increases the risk for site-specific cancers, based on the evaluation of several studies that included diet scores assessed by food-frequency questionnaires and not merely fruit and vegetable consumption. We advocate the use of more comprehensive diet scores in any future research in this field. Physical activity was restricted to moderate and vigorous activities to be congruent with the existing evidence about the associations of the levels of physical activity with several types of cancer, which also motivated separate evaluation of sedentary behavior. There have been no previous studies of this nature, with both physical activity and sedentary behavior typically evaluated in the same measure and labeled as either physical activity or physical inactivity [19,20,22]. In addition, we suggest that for future research, it might be beneficial to pursue a more precise physical activity and sedentary behavior assessment, which could be achieved by using available technology (i.e., smartwatches).

There are several limitations to this study. First, cancer diagnosis was self-reported, so we chose not to perform analysis by cancer type, even though this would have added more detailed information regarding the health behaviors associated with different types of cancer. Current literature has already shown that different types of cancer have different associations with certain health behaviors. For instance, smoking is associated with lung cancer [9]; alcohol consumption with cancer in the oral cavity, pharynx, esophagus, colon/rectum, and liver [10]; an unhealthy diet with an increased risk of colon cancer [12]; low physical activity levels with an increased risk for 13 cancers [13]; a high BMI with six cancers [14]; and a more sedentary behavior with colorectal and endometrial cancer [15]. If the quality of our cancer data can be improved in the future, we will seek to perform separate analyses for each cancer type, and will allow for more accurate classifications based on health behaviors. Second, we excluded self-reported skin cancer because it could introduce bias through over-reporting [37]. Third, because health behaviors were also self-reported, participants might have answered the questionnaires based on the social expectations of data collectors, causing some respondent bias. Fourth, we only used six health behaviors that are commonly reported in the literature, and it is possible that other comorbidities, such as type 2 diabetes or cardiovascular disease, would have influenced the results. Future research might also benefit from the inclusion of more variables to gain a better understanding of the lifestyle differences between cancer survivors and those cancer-free. Finally, this was a retrospective study, and the findings will need to be validated in a prospective data set.

## 5. Conclusions

Our findings suggest that there are no key health behaviors to classify participants in the north of the Netherlands as cancer survivors or cancer-free, considering that we included cancer survivors of any type in the same group. Age was the most important predictor in our models. The linear and nonlinear machine-learning algorithms used in this study performed similarly in classifying participants as cancer survivors or cancer-free. Although irrelevant for classification, BMI and alcohol consumption were identified consistently as key health behaviors among cancer survivors. In future research, we will address the limitations of this study and validate the findings in a prospective data set.

## Figures and Tables

**Figure 1 cancers-13-02335-f001:**
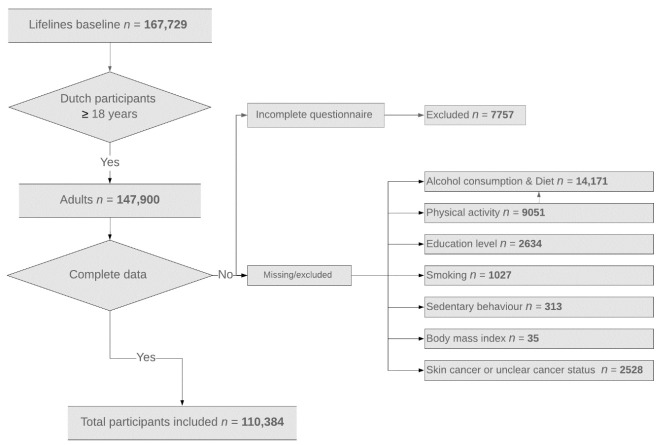
Study cohort selection based on health behaviors and socioeconomic factors.

**Figure 2 cancers-13-02335-f002:**
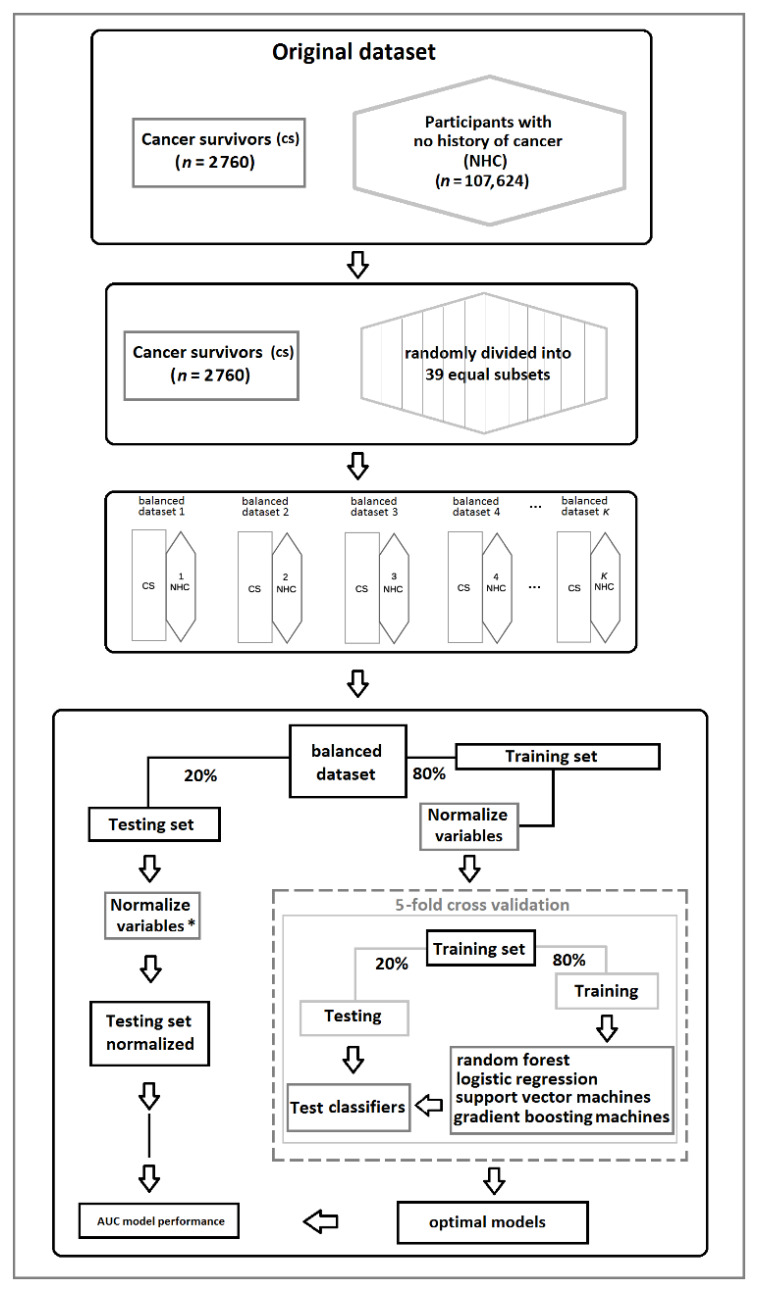
Overview of the procedure followed to reduce class imbalance (equalization strategy) in the Lifelines cohort. * Variables in the testing set were standardized using the mean and standard deviations from the training set.

**Table 1 cancers-13-02335-t001:** Baseline characteristics of participants stratified into cancer survivors, matched cancer-free controls, and cancer-free.

Variables	Cancer Survivors	Matched Cancer-Free Controls	All Participants without Cancer	*p*-Value
Participants	2760	2759	107,624	
Age, mean (SD)	57 (18)	57 (18)	44 (16)	*p* < 0.001
Sex, females (%)	1883 (68.2%)	1882 (68.2%)	62,910 (58.5%)	*p* < 0.001
Education level				
Low (%)	1209 (43.8%)	1179 (42.7%)	30,676 (28.5%)	
Medium (%)	862 (31.2%)	869 (31.5%)	43,107 (40.1%)	
High (%)	689 (25.0%)	711 (25.8%)	33,841 (31.4%)	*p* < 0.001
Time since cancer diagnosis				
≤5 years (%)	1153 (41.7%)			
>5 years (%)	1607 (58.3%)			*p* < 0.001
Body mass index	26.2 (5.20)	26.0 (5.10)	25.4 (5.20)	*p* < 0.001
Smoking g/day, mean (SD)	2.02 (5.69)	1.65 (4.85)	2.21 (5.75)	
Never (%)	1013 (36.7%)	1095 (39.7%)	50,624 (47.0%)	
Former (%)	1288 (46.7%)	1238 (44.9%)	35,067 (32.6%)	
Current (%)	459 (16.6%)	426 (15.4%)	21,933 (20.4%)	*p* < 0.001
Alcohol intake g/day	3.31 (9.35)	3.57 (9.24)	3.95 (9.46)	*p* < 0.001
Physical activity hrs/week	3.25 (5.75)	3.50 (5.58)	3.00 (5.00)	*p* < 0.001
Diet LLDS	26.00 (8.00)	26.00 (8.00)	24.00 (8.00)	*p* < 0.001
Sedentary behavior (TV hrs/day)	3.00 (1.61)	3.50 (1.50)	2.00 (1.50)	*p* < 0.001

Abbreviations: SD, standard deviation; hrs, hours; LLDS, Lifelines Diet Score; TV, television. Results are shown as median (interquartile range) unless otherwise specified.

**Table 2 cancers-13-02335-t002:** Overall performance of machine learning algorithms by AUCs for the 39 subsets and case–control analysis.

Scenarios	AUC 39 Subsets				AUC Case–Controls			
	Logistic Regression	Random Forest	Support Vector Machines	Gradient Boosting Machines	Logistic Regression	Random Forest	Support Vector Machines	Gradient Boosting Machines
All variables included * (95% CI).	0.75 ± 0.01	0.75 ± 0.01	0.76 ± 0.02	0.74 ± 0.01	0.52 ± 0.01	0.52 ± 0.01	0.55 ± 0.02	0.53 ± 0.01
- Excluding age (95% CI)	0.63 ± 0.01	0.63 ± 0.01	0.66 ± 0.01	0.65 ± 0.02	-	-	-	-
- Excluding age and sex (95% CI)	0.62 ± 0.01	0.63 ± 0.01	0.65 ± 0.01	0.64 ± 0.02	-	-	-	-
- Excluding age and education level (95% CI)	0.60 ± 0.01	0.62 ± 0.01	0.63 ± 0.01	0.61 ± 0.01	-	-	-	-

Abbreviation: AUC, area under the receiver operator curve. *All health behaviors and socioeconomic factors included.

**Table 3 cancers-13-02335-t003:** Consistency of variable importance in the random forest classifier by the MDG for every subanalysis.

Variables	All Variables	Health Behaviors *	Case–Control *
Age	100	-	-
Sex	7.65	-	-
Education level	6.03	-	-
Body Mass Index	56.44	100	100
Alcohol intake	54.04	99.42	99.15
Physical activity	45.87	83.23	84.93
Diet	43.77	73.95	76.93
Sedentary behavior	32.96	53.19	58.77
Smoking	13.27	12.84	24.30

The scale ranges from 1–100, where a number close to 100 means a more important variable in the analysis. The data show the consistency when including all variables, when including only health behaviors, and in the case–control analysis. * In these analyses, we included only health behaviors, therefore data for age, sex, and educational level are not shown.

## Data Availability

The data presented in this study are available on reasonable request from the corresponding author. The data are not publicly available due to data protection from the Lifelines Biobank (see https://www.lifelines.nl, accessed on 22 March 2021).

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
