# Peer review of "A Classification Approach for Cancer Survivors from Those Cancer-Free, Based on Health Behaviors: Analysis of the Lifelines Cohort"

_cancers, 2021, doi:10.3390/cancers13102335_

Round 1

Reviewer 1 Report

The authors of this paper focused on feature importance to distinguish cancer survivors from cancer-free participants. First, they trained a baseline model based on all risk factors using random forest and logistic regression. After that, predictive models were retrained by removing age or age and sex or age and educational level. However, they should write research contributions and explain what novelty is in their research.

The detailed comments are as follows:
1. The introduction should be extended by highlighting the contribution of the study. In addition, authors should clarify what problems are addressed in this study. For example, trying to improve predictive performance, solve class imbalance problem or identify key factors from the model, etc.
2. The experimental study is not enough. The authors need to use other machine learning algorithms, then need to explain risk factors. Because the main contribution of the presented work is not classification algorithm and they focused on the importance of risk factors. The conclusion has to be accurate.
3. Baseline characteristics of participants are provided in Table 1. Authors should provide significance analysis in each factor using a statistical test.
4. Their collected dataset has a class imbalance problem. They just randomly grouped major class (cancer-free participants) into 39 sub-sets that ratio was matched with minor class (cancer survivors). However, it is well known that the under-sampling technique has the biggest drawback to eliminate data reasoning loss of information. Why did not try to apply another synthetic over/under-sampling techniques? What is the specific reason to match the classes 1:1 using the under-sampling technique?
5. As shown in Figure 2, according to the sampling result, a total of 39 binary classification models corresponding to each subset. Each balanced group of the dataset may perform different performances, and their average is shown in Table 2. But, I interested to know variable importance scores in the random forest classifier as shown in Table 3. Did all models (39 models) produce the same variable importance scores? Please clarify it in Figure 2 or Subsection 3.4. Authors should show it by graphs.
6. Also, one of disadvantages is that very few variables have been used in this study, so it should be considered in further analysis.
7. The case-control study needs to be written in detail, and the Case-Control column in Table 3 is difficult to understand.
8. Why only AUC was used as an evaluation metric? Some evaluation metrics such as accuracy, precision, recall, and specificity can be used for performance evaluation. They have evaluated the performances of machine learning algorithms using the AUC metric. Another evaluation metrics should be added to prove your prediction results. Moreover, please use some visualization techniques to show your results, for instance, ROC curve, and variable importance analysis and so on.
9. As from a technical point of view, there is an important value of the paper if you experiment using various classification approaches of this paper.
10. There are some typos and English grammar issue too.

Reviewer 2 Report

It was a very interesting research, and I think it was a content that made us expect that similar research with higher accuracy will proceed in the future. I felt that the research should basically be adopted, but I would like to point out some points that should be considered. 

  1. It feels strange to include BMI in your health behaviour. Since I specialize in ethology, I have a strong sense of discomfort, but if there is no strong discomfort in oncology, it is okay to leave it as it is. If it is "improvable factor" or "improvable living factor", the discomfort may be lessened.
  2. Since the abbreviation of "AUC" is suddenly used in the Simple summary, I think it is better to describe it in a non-abbreviated form for the first time as in Abstract. 
  3. The last is a comment on future research. I think that if you can use physical fitness indicators (such as grip strength) rather than using self-reported physical activity and sitting behavior as indicators, your predictive power will greatly improve. We also expect the use of physical activity and sitting behavior by accelerometers and wearable activity meters in the future. 

Round 2

Reviewer 1 Report

Most of the problems has been fixed according to the comments.

However, as pointed out in comment 1, not only the contributions of this paper were not sufficiently described, and the differences between this paper and other previous studies were still insufficient.

Reviewer 2 Report

The part pointed out last time has also been corrected, and I got the impression that the quality of the paper has improved. It's a very minor point, but since "BMI" on line 111 is the first to appear, I thought it would be desirable to use a non-abbreviated notation. 
